# Examining the Sensitivity of Satellite-Derived Vegetation Indices to Plant Drought Stress in Grasslands in Poland

**DOI:** 10.3390/plants13162319

**Published:** 2024-08-20

**Authors:** Maciej Bartold, Konrad Wróblewski, Marcin Kluczek, Katarzyna Dąbrowska-Zielińska, Piotr Goliński

**Affiliations:** 1Remote Sensing Centre, Institute of Geodesy and Cartography, Modzelewskiego 27, 02-679 Warsaw, Poland; maciej.bartold@igik.edu.pl (M.B.); marcin.kluczek@igik.edu.pl (M.K.); katarzyna.dabrowska-zielinska@igik.edu.pl (K.D.-Z.); 2Department of Grassland and Natural Landscape Sciences, Poznań University of Life Sciences, Dojazd 11, 60-632 Poznań, Poland; piotr.golinski1@up.poznan.pl

**Keywords:** drought stress, grasslands, hydrothermal coefficient of selyaninov, plant response, satellite imagery, vegetation indices

## Abstract

In this study, the emphasis is on assessing how satellite-derived vegetation indices respond to drought stress characterized by meteorological observations. This study aimed to understand the dynamics of grassland vegetation and assess the impact of drought in the Wielkopolskie (PL41) and Podlaskie (PL84) regions of Poland. Spatial and temporal characteristics of grassland dynamics regarding drought occurrences from 2020 to 2023 were examined. Pearson correlation coefficients with standard errors were used to analyze vegetation indices, including NDVI, NDII, NDWI, and NDDI, in response to drought, characterized by the meteorological parameter the Hydrothermal Coefficient of Selyaninov (HTC), along with ground-based soil moisture measurements (SM). Among the vegetation indices studied, NDDI showed the strongest correlations with HTC at r = −0.75, R^2^ = 0.56, RMSE = 1.58, and SM at r = −0.82, R^2^ = 0.67, and RMSE = 16.33. The results indicated drought severity in 2023 within grassland fields in Wielkopolskie. Spatial–temporal analysis of NDDI revealed that approximately 50% of fields were at risk of drought during the initial decades of the growing season in 2023. Drought conditions intensified, notably in western Poland, while grasslands in northeastern Poland showed resilience to drought. These findings provide valuable insights for individual farmers through web and mobile applications, assisting in the development of strategies to mitigate the adverse effects of drought on grasslands and thereby reduce associated losses.

## 1. Introduction

Plant stress is a critical focus of environmental research, particularly in the context of climate change, ecosystem ecology, and conservation efforts [1,2]. Understanding how plants respond to various stressors such as drought, heat, cold, salinity, and pollution is essential for predicting how ecosystems will respond to changing environmental conditions [3]. In the face of climate change, extreme weather events such as heatwaves, droughts, and floods are becoming more frequent and severe. These events have had a significant impact on plant health and productivity. By studying the responses of plants to these stressors, researchers can better predict which species are most vulnerable and which may be more resilient in a changing climate. This knowledge is crucial for developing effective conservation strategies and mitigating the impacts of climate change on plant biodiversity. In grassland ecosystem ecology, plant stress research is particularly important because grasslands are highly sensitive to environmental changes [4,5].

Grassland ecosystems provide essential services such as carbon sequestration, water filtration, and habitat for a diverse range of species [6,7]. Understanding how different grassland species respond to stress can help researchers predict how these ecosystems will change in the future and develop strategies to conserve their biodiversity and ecosystem functions [8]. Additionally, plant stress research is essential for agriculture, as it helps farmers select crop varieties that are more resilient to environmental stressors, ultimately increasing crop yields and food security [9]. By studying plant stress responses, researchers can also develop new techniques for sustainable agriculture, such as drought-resistant crops and more efficient irrigation systems [10]. Therefore, plant stress plays a crucial role in environmental research, helping scientists understand how plants respond to changing environmental conditions and developing strategies to conserve biodiversity, protect ecosystems, and ensure food security in a changing climate [11].

Remote sensing has revolutionized the way we detect and monitor plant stress by providing valuable information on the spectral responses of plants to various stressors [12]. It involves the collection and interpretation of data from a distance, often using satellites, aircraft, or drones equipped with sensors capable of detecting different wavelengths of light. One of the key benefits of remote sensing for detecting plant stress is its ability to capture large-scale, spatially explicit information about plant health and vigor [13]. By measuring the spectral reflectance of plants across different wavelengths of light, remote sensing can provide valuable insights into the physiological and biochemical changes that occur in plants under stress. For example, stressed plants often exhibit changes in chlorophyll content, leaf water content, and canopy structure, which can alter their spectral reflectance properties. Remote sensing techniques such as multispectral and hyperspectral imaging can detect these changes by measuring the amount of light reflected by plants at specific wavelengths [14].

Multispectral imaging systems typically capture light in a few discrete bands, such as the visible and near-infrared spectra (NIR), allowing us to calculate vegetation indices such as the commonly used Normalized Difference Vegetation Index (NDVI). These indices provide valuable information about plant health and vigor, with low values indicating stressed or unhealthy vegetation [15]. This allows researchers to detect subtle changes in plant physiology and biochemistry associated with stress [16]. It is important to note, however, that this relationship does not hold for all vegetation indices. Other indices, such as the Enhanced Vegetation Index (EVI) or the Water Band Index (WBI), may respond differently to plant stress. In some cases, higher values could indicate stress or other physiological changes. Therefore, remote sensing science and techniques play a crucial role in detecting and monitoring plant stress by providing valuable information on the spectral responses of plants to various stressors [17]. By combining remote sensing data with ground-based observations and models, we can understand how plants respond to environmental stress and develop strategies to mitigate the impacts of stress on agricultural productivity, ecosystem health, and biodiversity conservation [18,19].

A widely used satellite data source comes from the Sentinel-2 satellites, which are highly popular due to their open-access policy, providing freely available data to the public [20,21]. The Sentinel-2 satellites provide high-resolution imagery at 10 m up to 60 m with a wide swath at 290 km, allowing for detailed observation and monitoring of changes in land cover [22], water bodies [23], vegetation, and ecosystems [24]. Secondly, the Sentinel-2 satellites capture imagery in 13 spectral bands, ranging from the visible to the shortwave infrared (SWIR), enabling researchers to analyze various vegetation indices and detect changes in plant health, biomass, and stress levels [25]. Additionally, the satellites revisit the same area on the Earth’s surface every 5 days with both satellites in operation, providing a frequent revisit time for regular monitoring of changes in vegetation, land cover, and environmental conditions. Lastly, the global coverage provided by the Sentinel-2 satellites allows researchers to monitor changes in vegetation, land cover, and environmental conditions across the entire planet.

A notable limitation of using Sentinel-2 for monitoring vegetation, including drought stress, is its optical sensor’s susceptibility to cloud cover, which can obscure the satellite’s view of the Earth’s surface [26]. This issue is particularly challenging in regions such as Poland, known for frequent cloud cover, where it can significantly hinder the continuity and reliability of the data [27,28]. Therefore, in our study, we used near-cloud-free images.

Our study aims to explore which commonly used vegetation indices retrieved from Sentinel-2 satellite imagery best capture changes in grassland vegetation caused by drought stress. We evaluated in situ measurements conducted during the growing season, as well as meteorological datasets, to identify conditions indicating drought. Then the meteorological and satellite imagery-based retrievals were examined to figure out the plant stress due to drought conditions. The manuscript proceeds with further analyses focusing on (1) the contribution of different vegetation indices from Sentinel-2 to determining plant drought stress, (2) the impact of utilizing high-resolution temporal satellite observations under frequent cloud coverage on the detection of drought severity in grasslands in Poland, and (3) the potential applications of our study’s findings for plant science and mapping drought stress using earth observation data. 

The innovative aspect of this research lies in its evaluation of various Sentinel-2 vegetation indices specifically tailored to detect plant stress, determining which most accurately reflect changes in grassland vegetation due to drought, and incorporating high-resolution temporal analysis under conditions of frequent cloud coverage, with implications for enhancing drought stress mapping using remote sensing data.

## 2. Materials and Methods

The following steps were taken to examine the sensitivity of satellite-derived vegetation indices to plant drought stress: (1) conducting field measurements in grasslands from 2020 to 2023; (2) creating a database of meteorological parameters to investigate drought conditions; (3) developing a database of vegetation indices primarily for identifying drought using Sentinel-2 imagery; (4) investigating and detecting drought through cross-verification of in situ and meteorological data with satellite information; and (5) culminating in the creation of illustrations that present spatiotemporal variations of drought intensity in Polish grasslands. We have included brief descriptions for context and employed color coding to visually distinguish between each step (Figure 1).

### 2.1. Study Areas

The study areas, which are highlighted with a red frame in Figure 2, are spread across different regions of the country, each characterized by unique natural conditions that include variations in terrain, water bodies, and vegetation structures. These areas vary from lowlands to hilly terrains, influencing the types of soil, their structure, and their properties. Furthermore, the sites range from regions with intense human activity to more pristine areas, adding to their diversity and making them valuable for research. In the Podlaskie region of northeastern Poland, coded as PL84 under NUTS 2 (NUTS2 is a classification level in the NUTS (Nomenclature of Territorial Units for Statistics) system of the European Union, used for regional analysis and allocation of structural funds), the landscape is mostly lowland but features a variety of terrain. The presence of rivers like the Biebrza and Bug enhances the region’s ecological diversity by supporting a rich mosaic of aquatic and wetland ecosystems. These river valleys are often humid, fostering unique ecological conditions that are conducive to the growth and development of the extensive grasslands [29].

Conversely, the Wielkopolskie region in western Poland, coded PL41 under NUTS 2, features a more balanced topography, predominantly composed of lowlands interspersed with some hilly areas. The Wielkopolskie Lowland, lying at the heart of this region, exemplifies a typical lowland terrain characterized by flat or gently rolling landscapes ideal for intensive farming [30]. This region is traversed by numerous rivers and streams, including the significant Warta River, which is a key hydrographic feature of the landscape.

Climatically, the regions span various climatic zones of Poland, leading to differences in temperature, precipitation, and vegetative periods. These climatic conditions directly influence soil development and its agricultural utility. Podlaskie experiences a moderately warm continental climate with a shorter growing season, with an average annual temperature of 6.5 °C and annual rainfall of 550 mm [31]. In contrast, Wielkopolskie is characterized by a warmer transitional climate with milder winters and warmer summers, where the average annual temperature is 7.5 °C and the total rainfall is 500 mm [32]. Both regions have experienced increases in temperatures and more frequent extreme weather events, such as droughts and heavy rainfall, in recent years [33]. In Wielkopolskie, key challenges include severe drought [34] and soil erosion [35].

The distribution and characteristics of grasslands across Poland present significant challenges for nature conservation [36]. Typically, these grassland patches are relatively small in size, increasing the risk of local species extinction. Additionally, they are often situated at considerable distances from one another, impeding the movement of populations between patches and hindering their ability to re-establish [30]. Analyses have shown that the proportion of permanent grasslands in the agricultural land structure in 2020 is significantly higher in the Podlaskie Voivodeship compared to Wielkopolskie, with proportions of 38.2% and 13.9%, respectively [30].

### 2.2. Field Measurements

Ground measurements were conducted in the grasslands recognized in the Podlaskie and Wielkopolskie voivodeships (Figure 1). A total of 46 field sites were selected for ground measurements, with 23 sites in each of the two voivodeships. Field sites were chosen to capture the variability and spatial distribution of the vegetation. In the Wielkopolskie region, the study area consisted of three fields smaller than one hectare, eighteen fields ranging from one to ten hectares, and two fields larger than ten hectares. Conversely, in the Podlaskie region, there were four fields smaller than one hectare and nineteen fields between one and ten hectares.

Ground measurements were synchronized with Sentinel-2 overpasses and were carried out every three to four weeks during the growing season from April to September. Table 1 lists the field campaign dates from 2020 to 2023. The frequency of ground measurements depended on cloud cover observations, the distances between specific fields, and the accessibility of convenient locations for easy access. During the ground campaigns, two biophysical parameters characterizing soil conditions and vegetation state were measured at the field sites: soil moisture (SM; measured to a depth of 15 cm) using the TRIME-PICO64 (Ettlingen, Germany) and grass height using the Electronic Bluetooth^®^ Plate Meter EC-20, supported by photo documentation.

### 2.3. Satellite Data Acquisition

The Sentinel-2 program, part of the Copernicus program, offers several advantages for environmental research and monitoring (https://sentinel.esa.int/web/sentinel/missions/sentinel-2 (accessed on 13 July 2024). The grassland areas in the Podlaskie region include three Sentinel-2 granules: 34UED, 34UFD, and 34UFE. Similarly, the grass fields in the Wielkopolskie region include three Sentinel-2 granules: 33UWT, 33UWU, and 33UXU. The satellite data are accessible from orbits 22, 122, 79, 36, and 136, allowing for image acquisition every 5–6 days. Sentinel-2A and Sentinel-2B satellite images at processing level 2A were automatically retrieved using Google Earth Engine (GEE), a cloud-based platform offering geospatial data, tools, and computational power for analyzing and visualizing satellite imagery and other geospatial data. Users can access and analyze the data using various programming languages, such as Python and JavaScript. GEE also provides a range of tools for data processing and analysis, including machine learning algorithms for image classification and time-series analysis [37]. For this study, we utilized the JavaScript API within the Earth Engine Code Editor. The climate in the two regions is moderately continental, characterized by a short growing season, prolonged snow and ice cover, and significant cloudiness. Consequently, we filtered the satellite data to include only those images with less than 10% cloud cover. Table 2 provides a comprehensive list of the selected Sentinel-2 images for the years 2020–2023 over the study areas.

Next, satellite meteorological datasets from ERA-5 Land, developed by the European Centre for Medium-Range Weather Forecasts (ECMWF), were utilized. ERA-5 Land is renowned as a top-tier tool for analyzing worldwide weather conditions and represents an enhanced iteration thanks to the incorporation of currently used technologies and refined algorithms [38]. The ERA5-Land dataset offers a spatial resolution of 0.1 degrees, equivalent to approximately 9 km × 9 km, providing detailed meteorological insights across diverse Earth regions. Spanning from 1950 to the present, this dataset facilitates the examination of climate trends over decades. Encompassing a wide range of meteorological variables, including air temperature, humidity, wind speed, atmospheric pressure, precipitation, and more, ERA5-Land provides a comprehensive toolkit for analysis. These parameters are accessible at different atmospheric levels. Demonstrating its versatility, ERA5-Land has proven invaluable in various scientific domains such as climate studies [38], land cover and vegetation seasonality [39], and weather forecasting [40]. In our study, we exploited ERA5-Land data, specifically daily averaged air temperature and daily total precipitation, to find out the sensitivity of remote sensing-based drought indices at grasslands.

### 2.4. Meteorological Drought Assessment

To investigate the meteorological conditions for grass development and detect drought, ERA-5 Land datasets comprising daily air temperature (Figure 3) and daily total precipitation (Figure 4) at grasslands during the growing season from 2020 to 2023 were utilized. It was observed that the distributions of average daily temperatures from 2020 to 2023 for grasslands in two distant voivodeships did not differ significantly. However, the thermal conditions for grass growth in both regions exhibited frequent temperature fluctuations, especially in 2022 and 2023, when temperature observations often surpassed the critical threshold of 20–25 degrees Celsius for average daily temperature. This threshold is considered the limit for thermal stress in plants under Polish climate conditions [41]. Additionally, concerning daily rainfall totals, distinct differences were observed in the temporal distribution of grasslands in both regions. In the Podlaskie Voivodeship, there were frequent instances of large daily rainfall totals exceeding 50 mm in 2020. Conversely, during a similar period in the Wielkopolskie Voivodeship, relatively consistent rainfall events were recorded, with daily totals not exceeding 40 mm. In subsequent years, daily rainfall totals generally ranged from 10 to 20 mm, depending on the observation period.

Next, the Hydrothermal Coefficient (HTC), also known as Selyaninov’s coefficient [42], was calculated to estimate meteorological drought conditions in grasslands. This widely used metric for drought monitoring in central and eastern European countries [43,44] characterizes the atmospheric moisture conditions that contribute to drought. The HTC is closely linked to the productivity of various grassland types [45]. Considering the widespread use of HTC in research, it was deemed an appropriate method for assessing the occurrence and intensity of drought.

The HTC combines air temperature and precipitation parameters over a specific period. HTC is calculated as the ratio of accumulated air temperature to accumulated precipitation over a given period. It is defined by the following Equation (1) [42]:(1)$$HTC=\frac{10\sum_{i=1}^{n}P_{i}}{\sum_{i=1}^{n}T_{i}}$$
where
*n*—length of the preceding period in days; *P_i_*—precipitation amount on the ith day (mm); *T_i_*—daily average of the air temperature on the *i*th day (°C).

This study adopted the widely accepted classification into nine HTC classes [46]. The HTC values are as follows: extremely dry (HTC < 0.4), very dry (HTC 0.4–0.8), dry (HTC 0.8–1.1), quite dry (HTC 1.1–1.4), optimum (HTC 1.4–1.7), quite humid (HTC 1.7–2.1), humid (HTC 2.1–2.6), very humid (HTC 2.6–3.0), and extremely humid (HTC > 3.0). In order to investigate the complexity and dynamics of meteorological drought for each analyzed moment within the growing season from 2015 to 2023, the median of the HTC index over the preceding 30 days (HTC30) was determined. The 30-day median HTC index is described as valuable because it smooths out short-term fluctuations and outliers, offering a more consistent and reliable measure of heat transfer performance [44].

### 2.5. Vegetation Indices Calculations

To assess how sensitive satellite-derived vegetation indices (VIs) are to plant stress, we selected and calculated VIs specifically designed for drought mapping using Sentinel-2 images of the study area. Initially, we identified the most commonly used spectral vegetation indices, which capture various aspects of plant growth, drought detection, and canopy water content. These VIs are detailed in Table 3.

The Normalized Difference Vegetation Index (NDVI) is a widely used measure of plant health and density, derived from the difference between the maximum absorption of red light and the maximum reflectance of near-infrared light. NDVI values range from −1 to 1, with higher values representing healthier and more abundant vegetation. This index is instrumental in monitoring vegetation and detecting changes in ecosystems and biodiversity. However, its accuracy can be influenced by atmospheric conditions, the soil background, and the structure of the plant canopy [51]. Moreover, when vegetation is dense and covers 100% of the ground, NDVI gives poor estimates of vegetation productivity because it is saturated and does not reflect the increase in biomass [52,53]. The Normalized Difference Infrared Index (NDII) specifies vegetation moisture and leaf water content. NDII values can range from −1 to 1, with higher values indicating greater water content in vegetation and lower values indicating less moisture. NDII is particularly useful for monitoring grassland productivity [54], detecting mowing frequency [55], and assessing plant hydric stress [56]. The Normalized Difference Water Index (NDWI), primarily developed by Gao in 1996 [49], is utilized to monitor changes in moisture levels and water content in grassland plants [57]. The last one, the Normalized Difference Drought Index (NDDI), was developed by Gu in 2007 [50] for assessing drought in grasslands. It is based on the relationships between two previously established indices: NDVI and NDWI. The value of the NDDI index increases with the severity of the drought. Assuming minimum drought criteria for NDVI < 0.5 and NDWI < 0.3, a drought condition occurs at NDDI > 0.25. The NDDI has been validated for evaluating drought conditions in grasslands in India [58] and Mongolia [59].

### 2.6. Estimating Drought Response to VIs

Considering that the NDVI and NDWI indices are included in the NDDI calculation and given that NDII is regarded as a less efficient indicator for monitoring drought conditions [57], we focused on investigating the sensitivity of the latest NDDI indicator values in relation to meteorological records. Additionally, ground measurements were used for cross-validation to assess the usefulness of the index that best characterizes drought severity. Furthermore, to verify drought severity, which is described within the value ranges of NDII and NDDI, we examined the significance of the relationships between them. We conducted a verification of the usefulness of vegetation indices through regression analysis in two stages. First, we assessed the extent to which changes in HTC values are reflected in the values of S-2 NDDI used as a drought indicator, and we examined their consistency with in situ soil moisture measurements taken at grassland sites from 2020 to 2023. The second stage focused on investigating drought severity by examining the relationship between NDDI and NDII. Finally, temporal variations of drought severity with S-2 NDDI at individual grassland parcels in the Wielkopolskie region during the growing season across the years 2020–2023 were studied. The performance statistics of the correlations were evaluated using various metrics, including the Pearson correlation coefficient (r), the coefficient of determination (R^2^), which is the square of the correlation coefficient, as well as the mean bias error (MBE), mean absolute error (MAE), and root mean square error (RMSE).

## 3. Results

The heatmaps (Figure 5) visualize temporal patterns of HTC30 during the growing seasons across years from 2015 to 2023 in Wielkopolskie and Podlaskie. The HTC30 is segmented into ten-day periods throughout the growing season, which spans from day 80 to day 300 of the year. The heatmaps illustrate a transition in meteorological conditions from wetter to drier over the years. Until the end of March (DOY 60–90), the data show predominantly “wet” and “extreme wet” conditions (blue shades), while during the growing season lasting until the end of September (DOY 90–270), there is a mix of all categories, indicating diverse conditions. In the most recent years, there has been a clear shift towards “dry” and “extremely dry” conditions (red shades). Specifically in Wielkopolskie, we noted frequent extreme dry and very dry conditions, i.e., 119 out of 216, which constitute 55% of all observations. This pattern suggests a long-term trend towards increased dryness, with significant seasonal variability within each year, reflecting the changing climate and its potential impact on the studied area. On the other hand, Podlaskie has experienced moderately dry and moderately wet conditions in recent years. The heatmaps provide a visual representation of changing conditions over time and highlight a shift towards drier conditions in recent years, which is critical for understanding long-term climatic trends and planning for future environmental and resource challenges.

Figure 6 shows the relationship between NDDI and HTC, characterized by a significant negative correlation coefficient r = −0.75 and R^2^ = 0.56. The errors are relatively low, i.e., MBE = 0.04, MAE = 1.28, and RMSE = 1.58. The blue trend line illustrates that as NDDI increases, HTC decreases, indicating that higher NDDI values, which signify greater drought severity, are associated with lower HTC values, reflecting drier conditions. The data points are predominantly concentrated within the lower NDDI range (0 to 1.5) and the higher HTC range (1 to 4), thereby substantiating the negative correlation. This strong relationship suggests that NDDI is an effective index for identifying areas with lower hydrothermal conditions, supporting its use for monitoring drought severity in the studied region.

To substantiate the hypothesis regarding the relationship between SM and NDDI, an investigation was conducted (Figure 7). The analysis revealed a strong negative correlation between the drought index and soil moisture at r = −0.82 and R^2^ = 0.67, indicating that higher NDDI values are associated with lower soil moisture levels. The regression line and the scatter of the points show that soil moisture decreases as NDDI increases, reinforcing the inverse relationship.

While NDII and NDDI are considered appropriate for determining drought severity, the relationship between NDII and NDDI was investigated as well (Figure 8). A blue exponential regression line runs through the data points, highlighting the general trend between NDII and NDDI at the level r = −0.71, R^2^ = 0.50, with a relatively low RMSE of 0.94. The line indicates that as NDDI values increase, NDII values tend to decrease, suggesting a negative correlation between the two indices. The plot also features annotations dividing the data into two regions: “stressed” and “severely stressed”. The “stressed” region, marked with an orange ellipse, contains a cluster of data points where NDII values are positive (ranging from 0 to 0.2) and NDDI values are relatively low (up to around 0.5). This region indicates conditions of moderate stress. In contrast, the “severely stressed” region, outlined with a red ellipse, encompasses data points where NDII values are lower (ranging from −0.1 to −0.3) and NDDI values are higher (from around 0.5 to 3.0). This area represents conditions of severe stress, where higher drought conditions correspond with lower NDII values. Overall, the scatter plot provides a visual representation of the negative correlation between NDII and NDDI, emphasizing the varying levels of stress indicated by different clusters of data points. NDDI offers several advantages over NDII for determining drought conditions. NDDI is specifically designed to measure drought by combining information from both vegetation and soil moisture content. This dual focus makes it a more targeted and direct indicator of drought stress. In contrast, NDII primarily measures vegetation moisture content, which can be influenced by a variety of factors other than drought, such as plant health, soil type, or recent rainfall. Consequently, while NDII provides valuable information about vegetation health and moisture, it does not exclusively indicate drought conditions. NDDI’s design to specifically detect drought stress makes it a more effective and reliable tool for identifying and assessing drought severity.

Figure 9 comprises four scatter plots, each representing the occurrence and intensity of drought conditions measured by NDDI across the years 2020–2023 at individual grassland fields in Wielkopolskie. Each plot showcases how NDDI values fluctuate over time, with each data point corresponding to observations from various fields or stations, indicated by distinct colored markers. In the 2020 plot, NDDI values generally remain low, with only a few instances where values exceed 1, indicating mild drought conditions. However, in 2021, there is a noticeable increase in NDDI values, especially around DOY 100–150, where several points exceed 3, signifying more severe drought conditions. The 2022 plot shows a broader distribution of higher NDDI values, particularly between DOY 100 and 200, with some points reaching up to 5, indicating very severe drought conditions during this period. The 2023 plot continues this trend, with numerous observations exceeding 2 and several reaching up to 5, particularly between DOY 100 and 200, suggesting persistent and severe drought conditions. Overall, the data indicates a trend of increasing drought severity over the four-year period, with 2022 and 2023 experiencing the most significant drought conditions. This pattern highlights the importance of continuous monitoring and analysis of NDDI values to manage and mitigate the impacts of drought on agricultural fields and other affected areas.

## 4. Discussion

This study demonstrated that satellite-derived indices for detecting plant stress offer new opportunities for continuous temporal and spatial monitoring of grassland vegetation health. The findings support the hypothesis that terrestrial ecosystems remain under stress due to hydrothermal conditions affecting the state and development of grass plants. The severity of the drought in 2023 was attributed to a significant increase in HTC, accompanied by higher daily air temperatures and a notable decrease in daily rainfall. Our study confirmed that the utilization of hydrothermal coefficients (HTC) enables us to explore spatio-temporal patterns of meteorological drought, thereby facilitating the detection of drought stress in grass plants. While HTC offers a comprehensive approach to quantifying the combined influence of temperature and moisture on vegetation, there are some limitations associated with their application. Their calculation entails complexities, relying on sophisticated models and algorithms, which can be computationally demanding [60] and may not uniformly capture the impact of temperature and moisture variations across different vegetation types and geographical regions, potentially introducing biases in interpretation [61]. Furthermore, the accuracy of HTC estimates hinges on the availability and quality of input data, including meteorological observations and remote sensing imagery [44].

In this study, well-validated Sentinel-2 surface reflectance products were used to determine vegetation indices (VIs) and analyze drought dynamics for 2020–2023. Shepherd [62] confirmed the quality of compositions from Sentinel-2 satellite images, demonstrating the use of improved cloud-free and composited daily, weekly, or monthly mosaics for regular land monitoring. Due to unfavorable weather conditions in Poland, where clouds are present on average 150 days per year [63], S-2 imaging scenes with less than 10% cloud coverage were taken into consideration. As shown in Table 2, the number of acquired images for the two distinct areas varied significantly across different years. For the Wielkopolskie region, nearly twice as many cloud-free or minimally cloudy satellite images were obtained. This spatial relationship confirms observations of spatial variability in the number of cloudy days in Poland, as documented by Sypniewska [63]. Moreover, a consistent and repeatable amount of annual Sentinel-2 scenes was collected for the years 2020–2023, regardless of geographic location. Similar variations in the availability of cloud-free or minimally cloud-covered satellite optical data have been noted in previous grassland studies conducted under Polish climate conditions [64,65]. This variability is an important factor to consider in future studies. Therefore, we recommend that future research investigating temporal patterns in remote sensing data should account for this factor and explicitly evaluate its impact on drought stress in grass plants.

Among the tested vegetation indices and relationships presented in Figure 6 and Figure 7, the NDDI has shown the strongest correlation with HTC and SM. The NDDI is highly applicable for detecting drought severity in grasslands due to its sensitivity to both vegetation and soil moisture conditions. It integrates NDVI and NDWI, leveraging the red and near-infrared spectra critical for NDVI and the shortwave infrared spectrum essential for NDWI. NDVI captures the photosynthetic activity of plants by distinguishing between healthy vegetation and stressed or sparse vegetation using red and NIR reflectance. NDWI, on the other hand, is sensitive to leaf water content, utilizing the SWIR spectrum to detect moisture levels in vegetation. By combining these indices, NDDI provides a comprehensive measure of both plant health and water stress, enabling more accurate and timely detection of drought conditions in grasslands. This dual sensitivity allows for better monitoring and management of drought impacts, ultimately aiding in the preservation of these critical ecosystems, as noted in the previous studies of Artikanur [66] and Patil [67]. Meteorological conditions significantly influence the values of NDDI, providing an added value in this study for detecting plant stress (Figure 9). The integration of meteorological data, such as temperature and precipitation, with NDDI allows for a more accurate assessment of drought severity and its impact on vegetation. This enhanced detection capability is crucial for understanding the extent of stress in plants, particularly in Polish grasslands under various climate zones, where the interplay between climatic factors and plant health is complex and dynamic [68,69,70]. 

The analysis of VIs revealed that plant drought stress can be detected and monitored using the NDDI provided at a 10-m spatial resolution from S-2 imagery. While many studies have analyzed drought in Poland using commonly used indices such as the Drought Information Satellite System (DISS) at a coarse 1000 m spatial resolution [71] and the Standardized Precipitation Evapotranspiration Index (SPEI) based on meteorological observations at a 0.25-degree spatial resolution [34], our approach offers local farmers the opportunity to investigate drought conditions with much greater precision at individual field levels. Farmers in the Wielkopolska region, which is strongly affected by soil erosion [35], should take particular interest in these types of analyses. Research examining drought frequency between 2001 and 2023, conducted using the DISS index methodology with satellite data from the Terra MODIS satellite [71], has highlighted that agricultural fields and grasslands in Wielkopolska are significantly more vulnerable to drought compared to the Podlaskie region (Figure 10). Therefore, the critical focus remains on linking adverse conditions for grass vegetation growth and providing precise spatial information at a resolution of 10 × 10 m, along with frequent temporal updates. These requirements are met by the satellite observations provided within the Earth Observation Copernicus S-2.

In the context of tracking drought severity with high temporal and spatial accuracy for individual farmers in Wielkopolskie, Figure 11 unveils new insights for utilizing S-2 NDDI observations. The figure illustrates the proportion of grassland fields endangered by drought in ten-day periods across four years. In 2023, drought risk peaked early in the year, notably between DOY 70 and 90, with over 50% of fields affected, a significantly higher proportion compared to the same period in other years. Another substantial increase in 2023 occurred around DOY 150 and 230, once again surpassing the other years. In contrast, 2021 exhibited consistently low drought risk throughout the year, while 2020 and 2022 showed moderate risk spikes at various periods. These data indicate that 2023 experienced more severe and frequent drought conditions in grassland fields, especially in the early and mid-year periods, underscoring a notable escalation in drought risk compared to previous years.

Examining drought sensitivity in agriculture has been the subject of numerous studies conducted by various researchers [72,73,74]. It has been proven that satellite-derived information provides valuable, repeatable, and near-real-time data. However, utilizing the Normalized Difference Drought Index (NDDI) with Sentinel-2 data for mapping drought severity comes with several uncertainties and limitations. One primary challenge is the variability in atmospheric conditions, such as cloud cover and aerosol presence, which can affect the accuracy of satellite imagery and its availability (Table 2). Additionally, the spatial resolution of Sentinel-2, while high, may still miss small-scale variations in drought conditions within heterogeneous landscapes. The temporal resolution also poses a limitation, as a ten-day period from five-day revisits might not capture rapid changes in soil moisture and vegetation stress. Furthermore, the NDDI relies on the accuracy of both NDVI and NDWI, which can be influenced by factors such as soil background, vegetation type, and phenological stage [51,57]. Calibration and validation of NDDI data against ground-based measurements are essential to ensure reliability but can be resource-intensive and geographically limited [67]. Thus, during field campaigns (Table 1), we collected in situ soil moisture data to confirm the reliability of the NDDI. These factors collectively introduce uncertainties in the assessment and mapping of drought severity, potentially affecting the precision of drought monitoring and decision-making processes [75].

## 5. Conclusions

The examination of vegetation indices to assess drought stress across various grassland regions in Poland has provided valuable insights into the intricate interplay between environmental factors and meteorological observations. This study emphasizes the critical role of effective grassland management in promoting agricultural sustainability amidst challenges posed by climate change, which often manifests in drought occurrences. The findings underscore the necessity for adaptive management strategies that prioritize resilience within agricultural systems. Distinct growth patterns in grasslands were observed, particularly concerning drought, as indicated by NDDI observations. While Wielkopolskie experienced increased drought occurrences in the first decades of the growing seasons from 2022 to 2023, Podlaskie exhibited stable conditions with minimal variations. These findings underscore the importance of considering local environmental factors when analyzing vegetation dynamics. 

Additionally, the examination of meteorological conditions revealed significant regional differences, particularly in temperature and precipitation. The year 2023 presented considerable challenges for vegetation development, characterized by unfavorable conditions across all study areas. These observations underscore the vulnerability of agricultural systems to climatic variability and the importance of adaptive management strategies. The use of the NDDI index for assessing drought response provided insights into distinguishing between wet and dry conditions. Significant variations in drought occurrences and the proportion of fields at risk were identified within Poland, influenced by climatic fluctuations and grassland management practices. The use of the NDDI index proved invaluable for assessing vegetation water content, offering a potential tool for monitoring and managing grassland resources.

The implications of these findings extend beyond academic research, holding practical significance for agricultural stakeholders. By enhancing our understanding of the relationship between environmental conditions and biomass production, farmers and land managers can implement more effective cultivation practices and grassland management strategies. This knowledge is particularly crucial given changing climatic conditions, where adaptive approaches are essential for ensuring the resilience and sustainability of agricultural systems. In conclusion, this study provides valuable insights into the complex dynamics of vegetation growth and response to drought across diverse agricultural landscapes. By elucidating the influence of environmental factors on agricultural outcomes, it lays the groundwork for informed decision-making and sustainable resource management in response to evolving climatic challenges.

## Figures and Tables

**Figure 1 plants-13-02319-f001:**
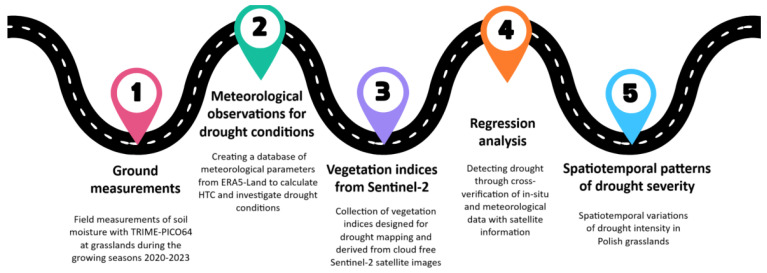
The process of examining the sensitivity of satellite-derived vegetation indices to plant drought stress at grasslands in Poland from 2020 to 2023.

**Figure 2 plants-13-02319-f002:**
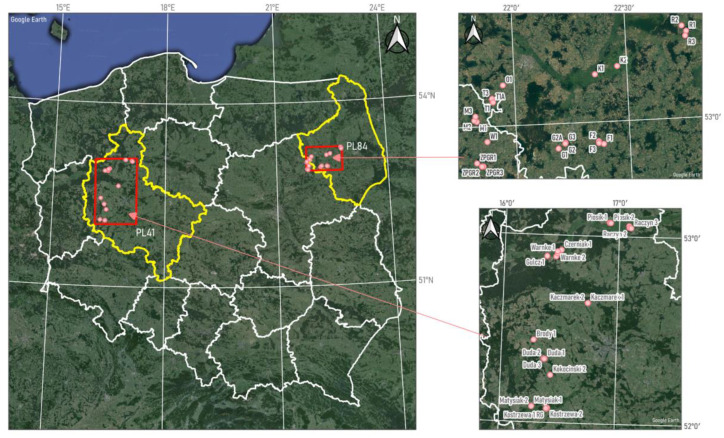
Location of field measurements conducted at grasslands highlighted by red dots in Wielkopolskie (PL41) and Podlaskie (PL84) provinces in Poland.

**Figure 3 plants-13-02319-f003:**
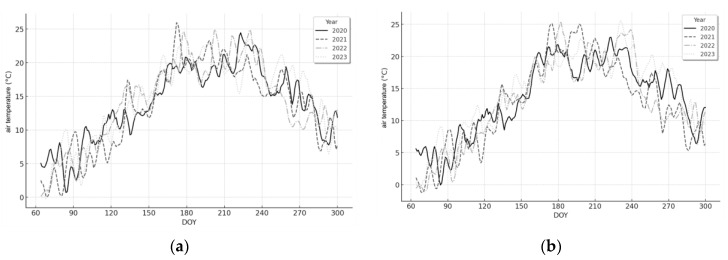
Average daily air temperature from March to October in 2020–2023 in the Wielkopolskie (**a**) and Podlaskie (**b**) voivodeships.

**Figure 4 plants-13-02319-f004:**
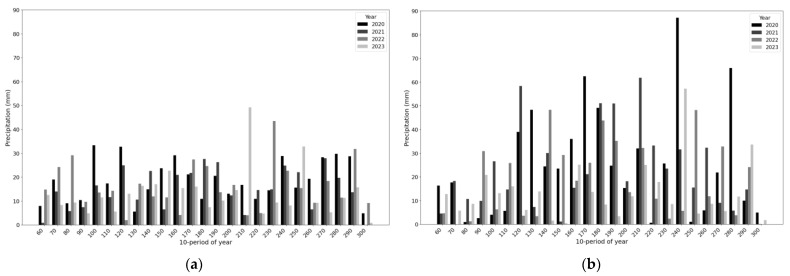
Daily total precipitation from March to October in 2020–2023 in the Wielkopolskie (**a**) and Podlaskie (**b**) voivodeships.

**Figure 5 plants-13-02319-f005:**
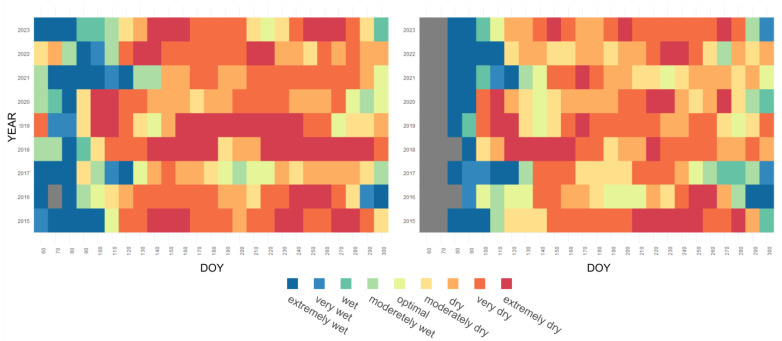
Temporal patterns of HTC30 at Wielkopolskie (**left**) and Podlaskie (**right**) regions during the growing season from March (DoY 60) until the end of October (DoY 300) in 2015–2023.

**Figure 6 plants-13-02319-f006:**
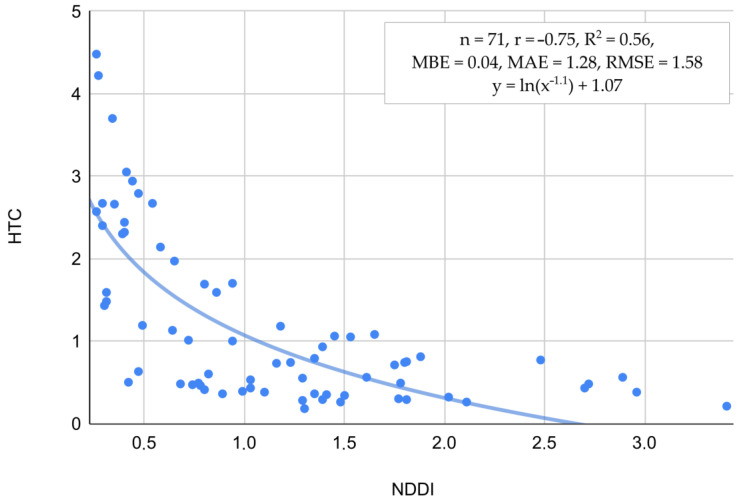
Relation between HTC and NDDI estimated across all satellite observations over grasslands recognized in this study.

**Figure 7 plants-13-02319-f007:**
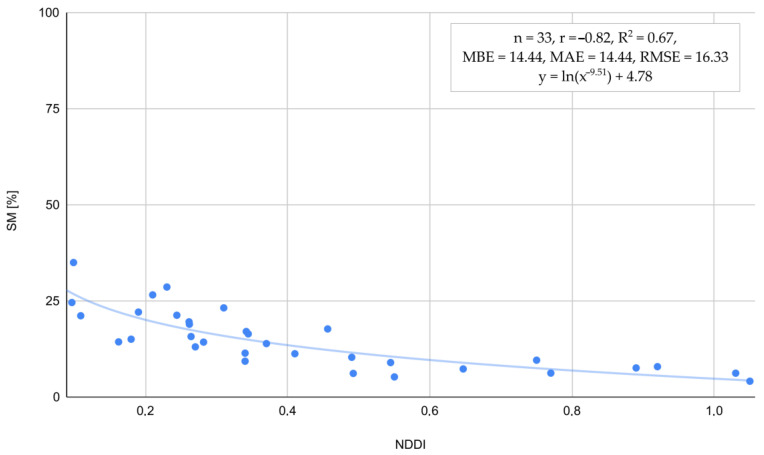
Relation between SM and NDDI estimated across all satellite observations over grasslands recognized in this study.

**Figure 8 plants-13-02319-f008:**
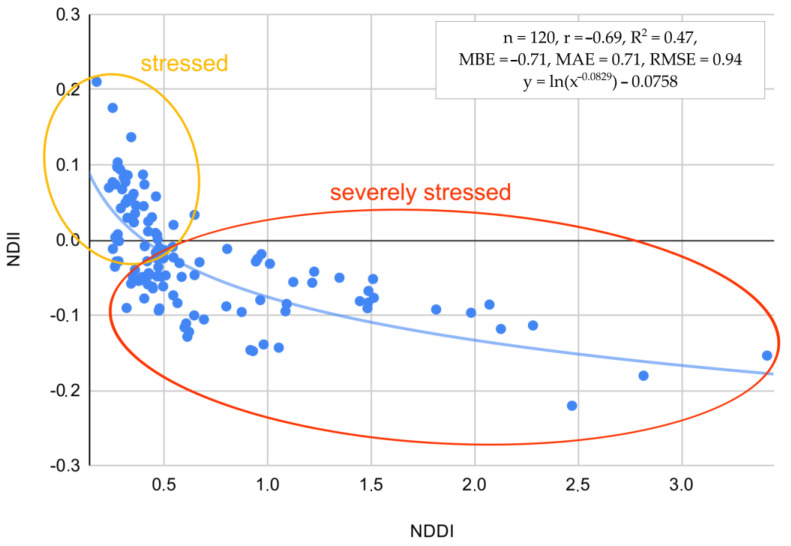
Relation between NDII and NDDI for classifying stressed and severely stressed plants to drought.

**Figure 9 plants-13-02319-f009:**
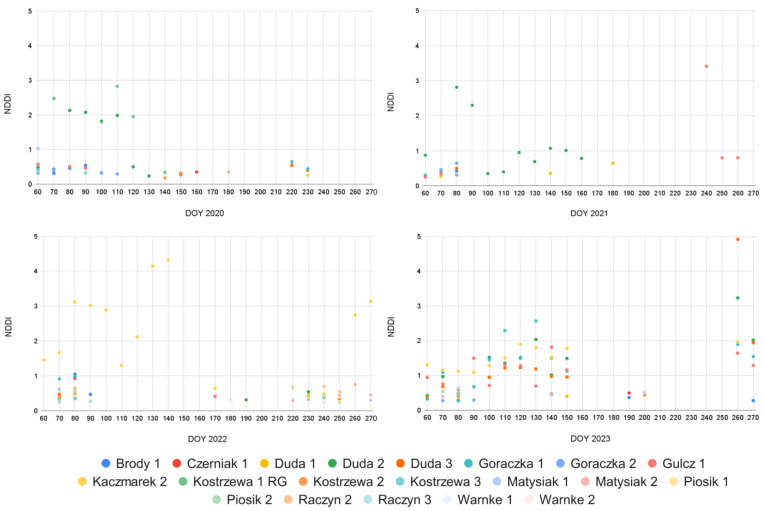
Temporal variations of S-2 NDDI at individual grassland parcels in the Wielkopolskie region during the growing season across the years 2020–2023.

**Figure 10 plants-13-02319-f010:**
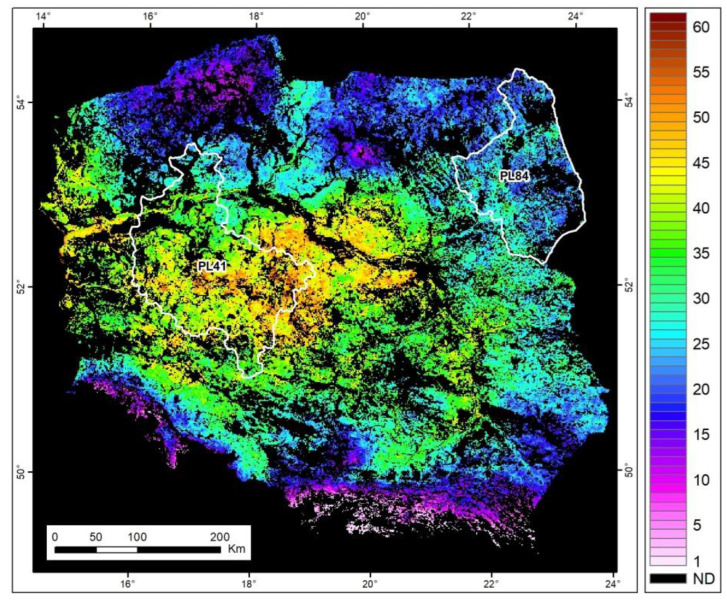
Frequency (%) of drought occurrences derived from satellite Terra MODIS observations across 2001–2023. White lines highlight our study areas.

**Figure 11 plants-13-02319-f011:**
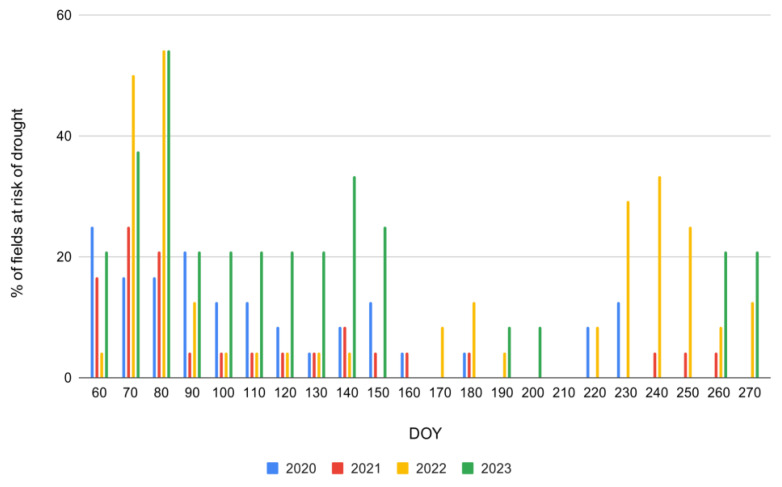
Frequency (%) of individual parcels in Wielkopolskie with recognized drought severity estimated from satellite Sentinel-2 NDDI observations across 2020–2023.

**Table 1 plants-13-02319-t001:** Ground measurement dates were conducted at Wielkopolskie and Podlaskie voivodeships across the years 2020–2023.

Year	Wielkopolskie	Podlaskie
2020	16.06, 22–23.06, 30.06, 15.07, 18.07, 06.08, 12–13.08, 23.08, 09–10.09, 17.09, 24.09	06–08.07, 28–29.07, 18–20.08, 30.09, 01.10
2021	24.04, 26.04, 08.05, 10.05, 21.05, 04–05.06, 17.06, 21.06, 02–03.07, 17.07, 29.07, 31.07, 02.08, 21–22.08, 04.09, 06.09, 19.09, 23.09	10–11.05, 09–10.06, 29–30.06, 26–27.07, 23–24.08
2022	22.04, 30.04, 07.05, 20.05, 22–23.05, 27.05, 18.06, 25.06, 08.07, 20.07, 24.07, 06–07.08, 25.08, 28–29.09	10–11.05, 22–23.06, 27–28.07
2023	29.04, 02.05, 12.05, 21.05, 17.06, 24.06, 09.07, 16.07, 27.07, 05.08, 13.08, 26–27.08, 10.09, 23–24.09	19–20.04, 24–25.05, 05–06.07

**Table 2 plants-13-02319-t002:** Number of S2 imaging scenes across years 2020–2023 used for the study.

Year	Wielkopolskie	Podlaskie
2020	108	60
2021	97	48
2022	107	45
2023	100	59

**Table 3 plants-13-02319-t003:** Spectral indices used in study.

Short Name	Full Name	Formula	Authors
NDVI	Normalized Difference Vegetation Index	$NDVI=\frac{R_{842}-R_{665}}{R_{842}+R_{665}}$	Rouse et al., 1973 [47]
NDII	Normalized Difference Infrared Index	$NDII=\frac{R_{842}-R_{1610}}{R_{842}+R_{1610}}$	Hardisky et al., 1983 [48]
NDWI	Normalized Difference Water Index	$NDWI=\frac{R_{860}-R_{1240}}{R_{860}+R_{1240}}$	Gao, 1996 [49]
NDDI	Normalized Difference Drought Index	$NDDI=\frac{NDVI-NDWI}{NDVI+NDWI}$	Gu et al., 2007 [50]

## Data Availability

The datasets used and analyzed during the current study are available from the corresponding author upon reasonable request.

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
