# Peer review of "Examining the Sensitivity of Satellite-Derived Vegetation Indices to Plant Drought Stress in Grasslands in Poland"

_plants, 2024, doi:10.3390/plants13162319_

Round 1

Reviewer 1 Report

Comments and Suggestions for Authors

Comments on plants-3134661-peer-review-v1

A general comment. It is not clear if the authors will treat, or have treated, the concept of “drought” as a severe form of “water stress”. Indices as NDVI address mainly water stress.

lines 28 to 69 This introductory text express basic, widely known concepts; perhaps this could be abbreviated.

line 69 it may be true, for NDVI, that lower values suggest plant stress, but this is not valid for all other indices. Other indices could be mentioned, as other investigations using several indices; the state-of-art is not addressed.

line 78 to write “The most commonly used satellite data comes from the Sentinel-2 satellites” is a bold statement. Better “A widely used satellite data source comes from the Sentinel-2 satellites”

line 93 this quite obvious limitation is common for all optical Remote Sensing satellites, not only Sentinel ones. Better to state that (almost) cloudless images were selected.

lines 111 and 116. The authors use the words “Earth Observation data”, with capital letters, in a confusing way. Is this a special service available to researchers? or is just a general expression of a Remote Sensing resource? it seems to be another way to write “Remote Sensing”.

line 137, caption of Fig. 2. The word “voivodeship” is very specialized, in use at Eastern Europe; suggest using “province” which is more international.

lines 144, 150 what is NUTS 2?

lines 167, 168 According to the referencing style being used in this paper, instead of “According to Szymura [36], the distribution and characteristics of grasslands across Poland present significant challenges for nature conservation”, write “The distribution and characteristics of grasslands across Poland present significant challenges for nature conservation [36].” Same at line 172, and line 256, and 274, and other places.

line 195. The dates of the satellite images could be informed, or instead provide an information on the superposition of dates of ground and orbital data.

line 217 cutting-edge is a relative term, should be avoided.

lines 216 to 229 this text on ERA5-Land is not necessary.

lines 268 to 271. Not necessary to show all the classes; anyway, it seems that there are nine classes, not ten.

lines 350, 356 this Pearson correlation coefficient is not particularly “strong”, it rather suggests a significant correlation.

line 362 SM stands for “soil moisture”, this should be indicated here or back in line 191

line 362 this sentence is confuse.

line 487 no need of “American”

Comments on the Quality of English Language

The paper is clearly written and it is easy to read. However, the authors adopted a style of employing short sentences that is sometimes tiresome. 

Author Response

Comment 1: “lines 28 to 69 This introductory text express basic, widely known concepts; perhaps this could be abbreviated.”

Response 1: Thank you for presenting your point of view. However, we chose not to shorten this section as we believe the information provided is crucial for the context of our research and helps to better understand the results presented.

Comment 2: “line 69 it may be true, for NDVI, that lower values suggest plant stress, but this is not valid for all other indices. Other indices could be mentioned, as other investigations using several indices; the state-of-art is not addressed.”

Response 2: We agree. We have taken the liberty of providing additional information regarding other illustrative examples of indicators utilized in the study, along with their observed behavior under varying conditions of vegetation status.

Comment 3: “line 78 to write “The most commonly used satellite data comes from the Sentinel-2 satellites” is a bold statement. Better “A widely used satellite data source comes from the Sentinel-2 satellites”

Response 3: We agree. We have corrected as suggested.

Comments 4: “line 93 this quite obvious limitation is common for all optical Remote Sensing satellites, not only Sentinel ones. Better to state that (almost) cloudless images were selected.”

Response 4: We agree. We have corrected as suggested

Comments 5: “lines 111 and 116. The authors use the words “Earth Observation data”, with capital letters, in a confusing way. Is this a special service available to researchers? or is just a general expression of a Remote Sensing resource? it seems to be another way to write “Remote Sensing”.”

Response 5: Thank you for the pertinent observation. We can consider earth observing data as a synonym for remote sensing. We have corrected according to the comment.

Comments 6: “line 137, caption of Fig. 2. The word “voivodeship” is very specialized, in use at Eastern Europe; suggest using “province” which is more international.”

Response 6: Thank you for your astute observation. We have corrected this issue in the text.

Comments 7: “lines 144, 150 what is NUTS 2?”

Response 7: NUTS2 is a classification level in the NUTS (Nomenclature of Territorial Units for Statistics) system of the European Union, used for regional analysis and allocation of structural funds, covering regions with a population between 800,000 and 3 million inhabitants. We supplemented the manuscript with these issues.

Comments 8: “lines 167, 168 According to the referencing style being used in this paper, instead of “According to Szymura [36], the distribution and characteristics of grasslands across Poland present significant challenges for nature conservation”, write “The distribution and characteristics of grasslands across Poland present significant challenges for nature conservation [36].” Same at line 172, and line 256, and 274, and other places.”

Response 8: Thank you very much for bringing this to our attention. We have corrected the citation method throughout the manuscript.

Comments 9: “line 195. The dates of the satellite images could be informed, or instead provide an information on the superposition of dates of ground and orbital data.”

Response 9: Thank you for your suggestion. We decided not to make changes. In our view, adding this information here could unnecessarily complicate the message.

Comments 10: “line 217 cutting-edge is a relative term, should be avoided.”

Response 10: Agree. We changed the formulation.

Comments 11: “lines 216 to 229 this text on ERA5-Land is not necessary.”

Response 11: Thank you for your suggestion. We decided to keep this section because the information on ERA5-Land is essential for a complete understanding of the context of our analysis and the reliability of the results obtained. We believe that including this information enriches the article and helps to better situate our research within the existing literature.

Comments 12: “lines 268 to 271. Not necessary to show all the classes; anyway, it seems that there are nine classes, not ten.”

Response 12: Thank you for your attention and insights. We have indeed identified nine classes and have made the necessary corrections in the text. After careful consideration, we have decided to leave the information regarding HTC classes as it is. We feel that including these details enhances the article and helps readers gain a deeper understanding of the presented results. For example, by reading the scatter plot it is possible to understand what the value of a given point means.

Comments 13: “lines 350, 356 this Pearson correlation coefficient is not particularly “strong”, it rather suggests a significant correlation.”

Response 13: Thank you for bringing this issue to our attention. We have corrected it in the manuscript.

Comments “14: line 362 SM stands for “soil moisture”, this should be indicated here or back in line 191”

Response 14: We added these issues in line 191

Comments 15: “line 362 this sentence is confuse.”

Response 15: We corrected the wording to make it clearer.

Comments 16: “line 487 no need of “American””

Response 16: We agree. We have removed this term from the text.

Reviewer 2 Report

Comments and Suggestions for Authors

Dear Authors,

The manuscript has potential for publication, it just needs some corrections and adjustments before final acceptance.

Author Response

Thank you very much for all your comments. We have taken your suggestions into account and made some changes to the abstract to make it more appealing. Thank you for your suggestions regarding the placement of certain sentences in a particular part of the manuscript. We have made the suggested corrections. We are also grateful for your helpful feedback on the technical aspects of the manuscript. We have corrected all related matters, including figure captions, tables, and the figures themselves. 

Reviewer 3 Report

Comments and Suggestions for Authors

I think that the use of remote sensing of vegetation for drought would have been enhanced by undertaking assessments of pasture condition and water status taken at the same time as soil moisture measures. Although some measures were taken (Table 1) they do not feature in the results or the discussion. This lack of pasture condition would make it difficult to determine the relevance of particular stages of drought as determined by NDDI or HTC30 to informed decision-making.

Drought has to be assessed within the context of local experience and management purpose and can be done in many different ways; these authors have chosen the NDDI and HTC30.

Despite this, I find that this research has produced a useful system for monitoring the role of climatic conditions on grasslands.

A couple of minor points:

L191 to what depth was soil moisture measured? It appears likely to be 30 cm, but plant roots will grow deeper than this.

In Figure 6 and Figure 7, the r values are given with SEs, in what appears to be for a curvilinear regression. The authors should include the regression formula in each Figure, including Figure 8.

Author Response

Comments 1: „L191 to what depth was soil moisture measured? It appears likely to be 30 cm, but plant roots will grow deeper than this.”

Response 1: I'm grateful for your helpful observation about the lack of this information. The soil moisture was measured at a depth of 15 cm. We have added this information in the text (line 194).

Comments 2: „In Figure 6 and Figure 7, the r values are given with SEs, in what appears to be for a curvilinear regression. The authors should include the regression formula in each Figure, including Figure 8.”

Response 2: Thank you for bringing this issue to our attention. We have supplemented all charts with this information.